## Are food and drink retailers within NHS venues adhering to NICE Quality standard 94 guidance on childhood obesity? A cross-sectional study of two large secondary care NHS hospitals in England

Alice James,[1] Laura Birch,[2] Peter Fletcher,[3] Sally Pearson,[4] Catherine Boyce,[4] Andy R Ness,[2] Julian P Hamilton-Shield,[2] Fiona E Lithander[2]

[1]Faculty of Health Sciences, University of Bristol, Bristol, UK
[2]NIHR Bristol Biomedical Research Centre (Nutrition Theme), University Hospitals Bristol NHS Foundation Trust and the University of Bristol, Bristol, UK
[3]Department of General and Old Age Medicine, Cheltenham General Hospital, Gloucestershire, UK
[4]Department of Clinical Strategy, Gloucestershire Hospitals NHS Foundation Trust, Gloucestershire, UK

**Correspondence to**
Dr Fiona E Lithander;
fiona.lithander@bristol.ac.uk

## ABSTRACT

**Objective** To assess whether the food and drink retail outlets in two major National Health Service (NHS) district general hospitals in England adhere to quality statements 1–3 of the UK National Institute for Health and Care Excellence (NICE) quality standard 94.

**Design** Cross-sectional, descriptive study to assess the food and drink options available in vending machines, restaurants, cafes and shops in two secondary care hospitals.

**Main outcome measures** Adherence to quality statement 1 whereby the food and drink items available in the vending machines were classified as either healthy or less healthy using the Nutrient Profiling Model (NPM). Compliance with quality statements 2 and 3 was assessed through the measurement of how clearly the shops, cafes and restaurants displayed nutrition information on menus, and the availability and prominent display of healthy food and drink options in retail outlets, respectively.

**Results** Adherence to quality statement 1 was poor. Of the 18 vending machines assessed, only 7 (39%) served both a healthy food and a healthy drink option. Neither hospital was compliant with quality statement 2 wherein nutritional information was not available on menus of food providers in either hospital. There was inconsistent compliance with quality standard 3 whereby healthy food and drink options were prominently displayed in the two main hospital restaurants, but all shops and cafes prioritised the display of unhealthy items.

**Conclusions** Neither hospital was consistently compliant with quality statements 1–3 of the NICE quality standard 94. Improving the availability of healthy foods and drinks while reducing the display and accessibility to less healthy options in NHS venues may improve family awareness of healthy alternatives. Making it easier for parents to direct their children to healthier choices is an ostensibly central component of our healthcare system.

## INTRODUCTION

Childhood obesity is a global health concern.[1] Almost 10% of 4–5-year-olds and 20% of

### Strengths and limitations of this study

► This is the first study to evaluate the consumer nutrition environment of two National Health Service hospitals in England, and to assess their compliance to recommendations of the National Institute for Health and Care Excellence quality standard 94.
► Only two hospitals in the same geographical area in England were assessed.
► The inclusion of a second assessor to provide an additional independent evaluation would have been beneficial to reduce possible bias.

10–11-year-olds in the UK are currently classified as obese[2 3] which is known to increase morbidity and mortality in later life through its association with cancer, cardiovascular disease and mental health disorders.[3 4] The doubling of obesity rates between the first and final years of primary school highlights early childhood as a high-risk period for obesogenic factors to take effect.[2 3] While overweight and obese children are more likely to become obese adults,[4] healthy weight adults who were obese during childhood have similar risk for metabolic diseases as those who maintained a healthy weight throughout life,[5] suggesting that childhood is an opportune time for obesity prevention.

There is increasing evidence that the consumer nutrition environment is an important determinant of dietary behaviour and obesity.[6–8] The consumer nutrition environment is one of four types of food environments and describes what an individual encounters when he or she enters a venue where food is purchased or consumed.[9] The consumer nutrition environment

**Table 1** NICE quality standard 94 'Obesity in children and young people: prevention and lifestyle weight management programmes'

|  | Quality statement | How it was assessed and how compliance was defined |
| --- | --- | --- |
| Quality statement 1 | Children and young people, and their parents or carers, using vending machines in local authority and NHS venues can buy healthy food and drink options. | Foods and drinks were classified as healthy or less healthy using NPM. A hospital was deemed compliant if it housed one vending machine which contained both one healthy food and one healthy drink. |
| Quality statement 2 | Children and young people, and their parents or carers, see details of nutritional information on menus at local authority and NHS venues. | A hospital was deemed compliant if the majority of its retails outlets displayed nutritional information on their menus. |
| Quality statement 3 | Children and young people, and their parents or carers, see healthy food and drink choices displayed prominently in local authority and NHS venues. | Food and drinks were defined as healthy if they were within the four food groups in the Eatwell Guide. A hospital was deemed compliant if the majority of its retail outlets prominently displayed healthy food and drink options. |

NHS, National Health Service; NICE, National Institute for Health and Care Excellence; NPM, Nutrient Profiling Model.

accounts for factors that influence food choice within shops such as availability, price, promotions, placement, variety, quality and nutrition information of food and drink,[7 10] and it is understood that consumers' dietary choices are affected by these factors.[7 11]

The UK National Health Service (NHS) recognises that venues such as hospitals are consumer nutrition environments given that hospitals are venues where food is purchased and consumed. Indeed the National Institute for Health and Care Excellence (NICE) quality standard 94 (table 1), which is entitled 'Obesity in children and young people: prevention and lifestyle weight management programmes',[4] identifies NHS venues as important settings in which to implement childhood obesity prevention strategies. The NICE quality standard 94 specifically refers to availability of healthy foods and drinks in vending machines, nutrition labelling of menus and prominent placement of healthy food and drink options in NHS venues.[4] Children may attend an NHS venue, including a hospital, as a patient or a visitor, and can be directly or indirectly influenced by the consumer nutrition environment within a hospital.

Studies conducted in Canada and the USA have assessed consumer nutrition environments in hospitals and have shown that food and drink retailers in both adult[12 13] and children's[14 15] hospitals offer unhealthy food and drink options. One of these studies[15] used a tool to assess the consumer nutrition environment by evaluating individual factors such as amount of nutritious food sold at cafeterias, the presence of fast food outlets, the amount of nutritious food alternatives and the availability of exercise programmes. The other three studies[12–14] used tools which allowed the creation of a composite score of the environment such as the Hospital Nutrition Environment Scan,[12 13] or the Nutritional Environment Measured Survey-cafeteria).[14] There are many other consumer nutrition environment tools in the literature that have been used in different venues which assess availability, price, variety and quality of fruit and vegetables,

advertising, product placement, price promotions and labelling.[7 16 17] Regardless of whether these tools assess individual factors or allow the creation of a composite score, it is noteworthy that few have been deemed reliable or valid. Moreover, most tools have been developed and used in the USA and Canada and have not been used outside North America.[7]

Not only is little known about the consumer nutrition environment in NHS hospitals in the UK but no tool exists which could comprehensively assess it. The current study addresses a gap in the literature by developing a consumer nutrition environment tool to measure the healthfulness of food outlets of NHS hospitals, and assesses this against NICE quality standard 94 which refers to the availability of healthy food and drinks in vending machines, nutrition labelling of menus, and the prominent placement of healthy food and drink options.

## METHODS
### Setting/hospitals
This study was conducted in two major district general hospitals in the South-West of England in July 2016. These two NHS hospitals were chosen as the lead author was studying in these hospitals during this period. Both hospitals offer adult inpatient care; paediatric, daytime access for assessment and outpatients are available in both, but only the larger hospital provides paediatric inpatient care. One hospital has approximately 680 beds in total, 8 vending machines and a total of 6 food retail outlets which comprise 1 restaurant, 3 cafes and 2 shops. The other has approximately 380 beds, 10 vending machines and 7 food retail outlets comprising 1 restaurant, 2 cafes and 3 shops. The two restaurants served hot and cold meals throughout the day. All shops sold sweet and savoury snack items, cold drinks and prepackaged sandwiches and salads. The only meals served by the cafes were sandwiches and salads in addition to confectioneryconfectionery, savoury snack items and hot drinks. All outlets

were accessible to both staff and patients. The only two outlets which appeared to differentially target staff rather than patients were the two cafes, one on each hospital site, within the respective education centres where staff training and meetings take place, and where several members of staff hold offices. Food and drink retailers in each hospital were evaluated against quality statements 1–3 of the NICE quality standard 94 (table 1). Data were collected by the lead author during a 2-week period in July 2016 where each vending machine and retail outlet was visited by her on one occasion only.

## Assessment tool

The Consumer Nutrition Environment Tool (C-NET) was developed by the lead author and used two different methods to measure adherence to quality standards 1–3 within the NICE quality statement 94.

Quality statement 1: 'Children and young people, and their parents or carers, using vending machines in local authority and NHS venues can buy healthy food and drink options'.

To measure adherence to quality statement 1, the user recorded each food and beverage item for sale in each of the vending machines. The nutrient composition for each item was then retrieved from the item packaging or from product nutrition information available online. The UK Department of Health (DH) Nutrient Profiling Model (NPM) was used to classify these food and drinks into two categories; less healthy and healthy, and details of this model and the scoring system have been published elsewhere.[18] To do this, the nutrient content of each food and beverage was assessed against a set of published criteria to determine whether it contains certain nutrients above or below particular thresholds. The NPM identifies food and drinks that are high in fat, salt or sugar and enables them to be differentiated into two categories, healthy or less healthy, based on their nutrient composition. NPM uses a scoring system where points are allocated on the basis of the nutrient content of 100 g of a food or beverage. Points are awarded for energy, saturated fat, total sugar and sodium ('A' nutrients) and fruit, vegetable and nut content, fibre and protein ('C' nutrients). The score for 'C' nutrients is subtracted from the 'A' nutrients score to give a final NPM score. Foods that score 4 points or more, and drinks which score 1 or more points are classified as 'less healthy' using NPM. An example is that of raw nuts which had a nutrient composition (per 100 g) as follows: 2656 kJ, 9.1 g saturated fat, 3.7 g total sugar, 0.02 g salt; this item scored 16 'A' points. On calculation of 'C' points for the raw nuts (per 100 g), they contained >80% fruit, vegetables or nuts, 6 g fibre and 15.8 g protein, giving it a total of 5 'C' points. The final score for this item was 1, hence this item was classified as healthy according to NPM. The number of vending machines within each NHS venue where each food and beverage was available was recorded and data collected were used to measure compliance with quality statement 1. NPM was developed by the Food Standards Agency in 2004–2005, and was subject to rigorous scientific scrutiny, extensive consultation and review. It is supported by the independent Scientific Advisory Committee on Nutrition and a wide range of nutrition experts.[19] NPM was introduced as mandatory in 2007 by the UK Office of Communications and DH to regulate food and drinks in the context of television advertising to children.

Quality statements 2 and 3, respectively: 'Children and young people, and their parents or carers, see details of nutritional information on menus at local authority and NHS venues' and 'Children and young people, and their parents or carers, see healthy food and drink choices displayed prominently in local authority and NHS venues'.

NPM was not used to measure compliance with quality statements 2 or 3; this was done using a more subjective assessment of the consumer nutrition environment by the lead author. To measure adherence to quality statements 2 and 3, she evaluated the quality of the consumer nutrition environment within each hospital restaurant, café or shop through answering a series of yes/no questions which centred around three main themes: the provision of nutrient information for meals on menus (quality statement 2), advertising and promotions (quality statement 3), and the prominent placing of healthy and unhealthy items in the retail outlets (quality statement 3) as outlined in box. The themes were derived directly from the two quality statements. For assessment against quality statement 2, for example, the lead author examined the retail outlets for the display of calories, fat, saturated fat and sugar on menus, as specified in the quality statement itself. Quality statement 2 was not relevant to the shops as they did not have menus.

For quality statement 3, the lead author's knowledge of the Eatwell Guide[20] was used whereby food within the food groups such as fruit, vegetables, dairy, wholegrain and high fibre foods, eggs, lean meat and pulses, were classified as healthy, and foods and drinks which are high in fat, salt and sugars such as biscuits, sweets, chocolate and sugar-sweetened beverages were classified as unhealthy. Quality statement 3 includes phrases such as 'can easily find' healthy food, and that these items were 'prominently displayed' and descriptions of these terms are outlined in box.

## Procedure

Compliance with quality statement 1 was observed if an NHS venue had a vending machine where at least one healthy food and at least one healthy beverage option were available for sale in the same machine. Compliance with quality statement 2 was observed if retail outlets provided information on energy, total fat, saturated fat, salt and sugar content of meals and snacks. Quality statement 2 states that listing ingredients and cooking methods constitutes an acceptable level of nutritional information if the information on energy, fat, saturated fat, salt and sugar content is not available. Drinks were not assessed as part of quality statement 2 since the rationale provided by NICE refers to food only. When nutrient information

Box    Consumer Nutrition Environment Tool (C-NET); the questions were asked to ascertain if retail outlets adhered to quality statements 2 and 3. Questions around nutritional information for hot and cold meals relate to quality statement 2. Questions around advertising and promotions, and the layout and prominent placing of healthy and unhealthy items relate to quality statement 3.

**Nutritional information for hot and cold meals**. *These questions relate to quality statement 2 'Children and young people, and their parents or carers, see details of nutritional information on menus at local authority and NHS venues'.*
Are comprehensive details on nutritional content available for sandwiches, salads and other cold and packaged meal options?
If this information is not provided, are the ingredients listed?
Are comprehensive details on nutritional content available for hot meal options?
If this information is not provided, are details provided about the ingredients and cooking methods?
Are posters or labels used to direct consumers to healthier meal options?
**Advertising and promotions**. *These questions relate to quality statement 3 'Children and young people, and their parents or carers, see healthy food and drink choices displayed prominently in local authority and NHS venues'.*
Are signs or other promotional materials used to advertise healthy options?
Are signs or other promotional materials used to promote unhealthy options?
Are there advertisements for unhealthy items that are clearly aimed at children? For example, the use of bright colours, television/cartoon characters and/or specific wording of advertisement?
Are price incentives (eg, reduced price offers) for healthy items clearly displayed (eg, using signs)?
Are price incentives (eg, reduced price offers) for unhealthy items clearly displayed (eg, displayed at payment area)?
**Layout and prominent placing of healthy and unhealthy items**. *These questions relate to quality statement 3 'Children and young people, and their parents or carers, see healthy food and drink choices displayed prominently in local authority and NHS venues'.*
Are healthy food and drink items promoted via prominent placing, for example, near the entrance to a shop or as part of a large/attractive display?
Are unhealthy food and drink options promoted via prominent placing?
Are unhealthy options displayed at the payment area or point of purchase?
Are unhealthy options displayed at an easy height for children to 'grab' (1 m height)?

was not clearly displayed, the lead author liaised with members of catering staff to identify whether this information would be available on request. With respect to quality statement 3, the lead author examined whether healthy and less healthy food and drinks were advertised outside the venues, whether advertising stalls containing food and drink items were placed nearest the door of the shop, café or restaurant, whether items were at a child's eye level which was classified as 1 m height, and whether items were displayed at the payment area. Compliance with quality statement 3 was observed if healthy food and

drink choices were displayed prominently in the retail outlets and poor compliance was deemed if unhealthy options were prominently displayed. Items were deemed to be prominently displayed if they were placed at the entrance to the retail outlet and hence visible to those walking past, beside queuing and payment areas, or if they were accompanied by signs advertising the product or detailing price promotions. Though price was not specifically mentioned in quality statements 1, 2 or 3, the price of vending machine items and the existence of price promotions were both assessed as part of this study. Quality statement 1 asks that venues ensure that children and their carers 'can buy' healthy items and quality statement 3 requires that healthy food and drink items are 'promoted'. Both statements thus require reasonable accessibility to healthy options. Accessibility applies to the physical display and number of items, and to the financial accessibility. The definition of compliance with the three quality statements was selected by the authors given the lack of guidance regarding what constituted compliance provided by NICE quality standard 94.

### Data analyses
Data on the NPM scored are presented as mean (SD) and range. Qualitative data were initially coded and collated into themes by the lead author. Interpretation of these data and the identification of themes was reviewed and discussed by multiple authors (AJ, LB, FEL) throughout the process.

### RESULTS
Two restaurants, 5 cafes, 5 shops and 18 vending machines were included in the analyses across both hospital sites. Data were originally collected from 29 vending machines but 11 were excluded from analyses. Ten vending machines, which sold hot drinks only, were excluded because nutrition information was not clearly displayed nor was it available online or from the supplier when requested by the lead author. One vending machine, which sold only frozen items such as burgers, was also excluded because this machine was removed from the hospital before the end of the data collection period.

Quality statement 1: 'Children and young people, and their parents or carers, using vending machines in local authority and NHS venues can buy healthy food and drink options'.

Of the 18 vending machines, 7 (39%) offered both a healthy food and a healthy drink (table 2). Fifty-five different items were on sale across all 18 vending machines, 40 of which were food and 15 were drinks. Where the same drink was served in a 330 mL can or in a larger 500 mL bottle, this was classified as two separate drinks. When the 40 food items alone were assessed using NPM, the mean (SD) nutrient profiling (NP) score was 18.3, (9.3). Only 4 of the 40 (10%) food items were classified as healthy with an NP score of less than 4; these items were baked crisps, a packet of dried fruit and nuts, a

**Table 2** Mean (SD) Nutrient Profiling Model Scores for all 40 food items and 15 drinks found in the vending machines; data include healthy and less healthy food and drinks. A food item is classified as less healthy where it scores ≥4 points. A drink is classified as less healthy where it scores ≥1 point

| | Mean | SD | Range |
|---|---|---|---|
| Crisps, n=10 | 10.3 | 4.3 | 1, 17 |
| Chocolate, n=19 | 25.6 | 1.3 | 23, 27 |
| Sweets, n=2 | 15.5 | 0.7 | 15, 16 |
| Sweet and savoury biscuits, n=4 | 19.8 | 6.7 | 10, 25 |
| Dried fruit and/or nuts, n=5 | 6.4 | 12.8 | −8, 25 |
| Drinks, n=15 | 0.5 | 1.6 | −4, 2 |

packet of raw nuts and a muesli bar (mean (SD) NP score −1.3 (4.5)). When the other 36 (90%) food items were analysed according to NPM, the mean (SD) NP score was 20.5 (6.8), significantly greater than the cut-off for the less healthy classification of 4 points or more. The four healthy foods items were priced equivalently to similar less healthy items available in the vending machines.

When the drinks were assessed, 8 of the 15 (53%) available were classified as healthy (NP score −0.5 (SD 1.4)) using the NPM cut-off of 1 point or more, and the remainder classified as less healthy (NP score 1.7 SD (0.5)). The drinks that were classified as healthy were an orange flavoured sugar-sweetened beverage, concentrated orange juice, bottled water and sugar-free cola drinks.

### Quality statement 2: 'Children and young people, and their parents or carers, see details of nutritional information on menus at local authority and NHS venues'

Neither hospital was compliant with quality statement 2. When the two restaurants were analysed against quality statement 2, it was found that neither displayed nutrition information on their menus for hot or cold meals. When the lead author asked the retail staff, she was informed that this information was available only for cold fillings offered with jacket potatoes and for a proportion of the filled sandwiches on sale. Quality statement 2 states that if the nutrient content of a recipe is unavailable to consumers, the ingredients and cooking methods should be available. While this information was not displayed on menus, catering staff advised that this information would be available on request, though it was not specifically requested by the lead author as part of this study. Though not displayed on menus, a range of sandwiches and salads that were made onsite were available in the restaurants. Nutritional information detailing total energy, fat, saturated fat, sugar and salt content was displayed on the labels of these items.

None of the cafes provided nutrient information on their menus, meaning that all five cafes were not compliant with quality statement 2. However, three of the cafes offered prepackaged sandwiches and comprehensive nutrition information was available on the packaging of these. The remaining two cafes sold sandwiches, baguettes and salads made by a local catering company. Catering staff in these cafes advised that specific nutrition information was not available for these items but that details of ingredients were available on request, though the lead author did not specifically request this information.

### Quality statement 3: 'Children and young people, and their parents or carers, see healthy food and drink choices displayed prominently in local authority and NHS venues'

There was inconsistent compliance to quality statement 3 in both hospitals (table 3). When the two restaurants were analysed, adherence to quality statement 3 varied. One of the restaurants housed a stall inside the entrance, which sold fresh fruit, vegetables and local produce, and was labelled as a 'Farm Shop'. The same restaurant also sold steamed potatoes and these, in addition to fresh vegetables, were clearly advertised as a cheaper option than a portion of chips. In the second restaurant, there were several signs which advertised 'Healthier Options'. This included a 'lighter breakfast' which comprised grapefruit segments, natural yoghurt, a bowl of breakfast cereal with milk, a pastry and a cheese portion, both of which are less healthy options. Healthier snack options such as fresh fruit were prominently displayed on service counters and by the payment area. All five cafes demonstrated good adherence to quality statement 3. The balance of healthy and less healthy items on display in the most prominent areas such as the queuing or payment area was equal and there was less evidence of advertising and price promotions on less healthy items than in the shops. Fresh fruit was available beside the payment areas in four of the five cafes. However baked goods such as muffins, scones, cookies and cakes were displayed in glass cabinets or on the counter surface next to the queuing and payment area in all five cafes.

Healthy options made up approximately 25% of the food and beverage items available in the shops and these items were displayed in less prominent locations such as at the back of the shop or on low-lying shelves. Posters were used in all of the hospital shops and cafes to advertise price promotions on less healthy options including signs placed outside shops and cafes to advertise items such as hot dogs and ice cream. Cakes made onsite were displayed next to the payment area in two of the shops, and less healthy options including muffins and chocolate bars were displayed at the payment areas in all shops and cafes. Price promotions were advertised for several of these less healthy products displayed at the payment area.

The layout of the unhealthy items in many of the shops appeared to target children. The 'pic n mix' sweet stalls in both of the largest shops had prominent positions near the entrance and products such as colourful sugar-sweetened iced drinks, lollipops and chocolate eggs containing small toys were displayed at the payment area or at a child's eye level. There was a display in one shop with a sign which read 'Big Kids Sweet Zone' offering 35 different sweets

**Table 3** Observed activities which shops, cafes and restaurants engaged in against which compliance to quality statement 3 was measured. The number in parentheses is the number of shops or cafes or restaurants that engaged in this activity

Quality statement 3: 'Children and young people, and their parents or carers, see healthy food and drink choices displayed prominently in local authority and NHS venues'

| | Compliant | Non-compliant |
|---|---|---|
| Shops, n=5 | Advertising and promotions<br>▲ 'Meal-deal' promotion includes sugar-free drink options (5)<br>▲ Poster advertising price promotions on fresh fruit and dried fruit (2) | Advertising and promotions<br>▲ Fruit available was not reflective of the fruit advertised on the poster (2)<br>▲ 'Big Kids Sweet Zone' offering 35 different products at 'pocket money' prices (1)<br>▲ Posters advertising price promotions on unhealthy options including share-size confectionery, sugar-sweetened beverages, crisps, 'coffee & muffin deal' hot dogs and ice cream (5)<br>▲ 'Low saturated fat' sandwich range available but not advertised (2)<br>▲ 'Meal-deal' poster advertising sandwich, crisps or piece of fruit plus a drink although the fruit option was unavailable in some shops (3)<br>Layout and prominent placing of healthy and unhealthy items<br>▲ Fresh fruit stall located at the back of the shop (1)<br>▲ Bruised fruit (1)<br>▲ First visible items on entry were pic 'n' mix stalls, confectionery and crisps (5)<br>▲ Floor-to-ceiling confectionery stall adjacent to the entrance (1)<br>▲ Multiple rows of confectionery at 1m height (5)<br>▲ Ice cream freezer at the entrance and visible from main hospital concourse (2)<br>▲ Home-made cakes prominently displayed at payment areas (2)<br>▲ Price promotions on unhealthy snack items prominently displayed at payment areas (2) |
| Cafes, n=5 | Advertising and promotions<br>▲ Poster advertising healthier wraps (roast chicken rainbow/sweet potato/falafel) (1)<br>▲ Skimmed/semiskimmed/soya milk advertised as options for drinks and porridge (1)<br>▲ Poster and tabletop advertisements for fruit smoothies containing 100% fruit in 500mL portions stating '1 of your 5 a-day' (1)<br>Layout and prominent placing of healthy and unhealthy items<br>▲ 'Healthier range' wraps/sandwiches/salads available and clearly labelled (1)<br>▲ Healthy breakfast options clearly displayed. Options included porridge pots, fruit salad and yoghurt (2)<br>▲ Fresh fruit salad tubs prominently displayed in glass display cabinet (1)<br>▲ Salads served without dressing. Self-service dressings available (1)<br>▲ Prices available for soup with or without a bread roll (1)<br>▲ Two healthier snack options prominently displayed which were cereal bars and healthier biscuits (2)<br>▲ Fresh fruit prominently displayed although no price information was available (1)<br>▲ Baked crisps available (2) | Advertising and promotions<br>▲ Price promotions on baked goods such as brownies displayed prominently next to the payment area (1)<br>▲ Poster advertisement in main hospital foyer outside café for 'double chocolate cookie mocha creamy cooler' (1)<br>▲ Poster advertisement for 'Product of the month' which were cookies (1)<br>Layout and prominent placing of healthy and unhealthy items<br>▲ Cakes and other baked goods prominently displayed at service and payment areas (5)<br>▲ No healthy snacks or fruit available (1)<br>▲ Crisps and salted nuts prominently displayed next to café entrance and in front of healthier snack items (2)<br>▲ Large chocolate bar selection available (5) |
| Restaurants, n=2 | Advertising and promotions<br>▲ Multiple healthy options advertised in prominent positions including cold breakfast bar, fresh fruit and steamed potatoes (2)<br>▲ Healthier 'side options' advertised using poster entitled 'It's better for you' on the service counter (1)<br>▲ Poster advertising 'now serving (Brand Name) (no added sugar) in our restaurant' (1)<br>▲ Poster advertising 100% fruit smoothies (2)<br>▲ Poster advertising 'lighter breakfast' options including cereal, yoghurt, fruit, mixed nuts (1)<br>Layout and prominent placing of healthy and unhealthy items<br>▲ First visible stall on entry is a farm stall displaying fresh fruit and vegetables (1)<br>▲ Choice of four packaged salads available in addition to sandwiches (1)<br>▲ Fruit and healthier snack options displayed near payment area (2)<br>▲ Healthier and cheaper 'side options' available which include potatoes and vegetable options with clear, comparable prices (2) | Advertising and promotions<br>▲ Price promotion on home-made cookies with self-service tongs at payment areas (1)<br>Layout and prominent placing of healthy and unhealthy items<br>▲ Self-service salad bar no longer in use (1)<br>▲ 'Snack' table in central location near payment areas predominantly serving baked goods (1)<br>▲ Snack bar and ice cream freezer located next to payment area and cutlery collection area (1)<br>▲ Menu displayed by restaurant entrance on stall sponsored by Coca-Cola and states 'Coca-Cola – complete your meal' (1) |

all priced at less than £1. Both of the largest shops used posters to advertise a price deal on fresh fruit, however fruit stalls were not in a prominent location and were poorly stocked.

## DISCUSSION

This study is the first to describe the consumer nutrition environment of two NHS hospitals in England and to assess their compliance with NICE quality standard 94. We found that food and drink retailers in these two hospitals demonstrated poor compliance with this quality standard. Only 39% of vending machines across both hospitals served both one healthy food and one healthy drink option indicating poor compliance with quality statement 1. Moreover, 90% (36/40) of the food items in the vending machines were classified as less healthy suggesting that the consumer may have difficulty identifying and locating the 10% of items classified as healthy. Of the drink items available 53% (8/15) were classified as unhealthy. It was found that the two hospitals were not compliant with quality statement 2, which refers to the availability of nutritional information at the point of choosing food or drink. Compliance was variable in relation to quality statement 3, where restaurants engaged in various activities in the display of healthy options yet all cafes and shops favoured the prominent display and advertising of unhealthy foods and drinks.

Hospitals have a role to play in advocating for healthy lifestyle and good nutrition.[21] [22] Choice architecture describes the concept that behaviour could be changed in anticipated ways by changing the environments where people make choices.[23] [24] The alteration of microenvironments, which are settings where people may congregate for a specific purpose,[25] including hospitals, may be one approach to encourage healthier dietary behaviour, and studies have shown that minor changes in accessibility to food can decrease food intake.[26] The consumer nutrition environment in 14 children's hospitals in California was assessed and it has been suggested that nutrition intervention is needed to improve the availability of healthy food and beverage options.[14] The authors of that research suggest that inexpensive interventions could be used such as providing nutrient information and introducing signage that promotes healthy choices.[14] McDonald and coworkers[15] reported that university-affiliated children's hospitals in Canada and the USA provide a suboptimal health environment, and hypothesise that a reliance on revenue from outlets which provide less nutritious foods may be a factor. Such results are not restricted to children's hospitals however. Winston and coworkers[12] [13] described the nutrition environment of 39 hospitals in the USA, and found that the consumer nutrition environment was poor and suggested that dietary interventions are justified in health settings.

The current study found that only 39% of vending machines provided both a healthy food and a healthy drink, and yet this result must be interpreted with caution. The most widely available healthy food item was baked crisps which were found in 33% vending machines, and when presented in the same vending machine as a healthy drink such as bottled water or a sugar-free drink, the machine, and by extension the hospital, was classified as compliant with quality statement 1. It is known that nutrient profile schemes have become drivers for product reformulation[18] and the baked crisps, and indeed the orange flavoured sugar-sweetened beverage which was classified as healthy, may have been reformulated to meet the NPM criteria. It is noteworthy that both of these items are classified as unhealthy according to the Eatwell Guide[20] where they are both in the category of foods which are high in fat, salt and sugar yet are classified as healthy when the NPM was used.

It is well accepted that nutrition information at the point of purchase can influence food choice.[27] Consumers underestimate by two to four times the saturated fat, energy and sodium content of restaurant foods,[27] yet providing accurate point-of-sale nutrition information is known to improve consumer choice.[27] None of the restaurants, cafes or shops examined in the current study were compliant with quality statement 2, which states that nutrition information should be available for consumers at the point of choosing food and drink options. It is a hospital's duty to empower consumers with the information required to make an informed choice.[28] When asked, staff suggested that nutrition information would be provided to consumers on request, yet the format of this information, exactly when it would be provided, and how user-friendly it would be, is unknown.

The availability and accessibility of unhealthy foods have been identified as risk factors for overeating[29] and it is understood that the prominence of food and drinks displays can influence consumer choice.[30] [31] In the current study, the restaurants demonstrated variable compliance to quality statement 3 which states that healthy food and drink options are displayed prominently. Evidence of good practice included one restaurant which advertised fresh fruit and vegetables at competitive prices, while the other advertised what was referred to as a 'lighter breakfast'. This option, however, also included a pastry and cheese portion, both of which are known to be high in fat, saturated fat and salt. The five shops and five cafes all prioritised the prominent display of unhealthy options, and while fruit was available beside the payment area in four of the five cafes although poorly stocked, so too were unhealthy baked goods which were displayed by the payment area in all five cafes. Research has shown that unhealthy food and drink items are difficult for consumers to avoid in supermarkets, as they take up more shelf space than healthy items such as fruit and vegetables, and are more often displayed at payment areas, as seen in the current study.[32–34] Moreover, the placement of healthy food items at a payment area can lead to substantial positive impact on sales of these products.[35] In addition, both of the largest shops in the current study displayed an array of sweets, chocolate and sugar-sweetened drinks at child's

eye level, and evidence suggests that placing products on shelves at eye level positively influences sales.[30]

The C-NET used in the current study was developed by the lead author of this study. While this tool has not been validated, it allowed both the objective collection of data using NPM, and the subjective assessment of marketing practices that contribute to food purchases using a series of questions (box 1). Other studies which have assessed food retailers in hospitals have used a variety of methods; some relied on telephone interviews with cafeteria directors which may have biased the data collected[15] while others used a tool such as the University of Pennsylvania Nutrition Environment Measures Scale. However, no equivalent British tool was found and thus the lead author developed C-NET. This model was selected for use in the current study as it is widely regarded as scientifically robust and effective in identifying less healthy items, and in practical terms, it is well established in the UK. There are other government endorsed NPMs available, such as the EU Pledge model and the WHO Europe model, but these use multiple categories and subcategories of foods which have raised concerns of ambiguity and additional complexity that may reduce the clarity and transparency of the models.[36] They also do not have the same track record of effective use in the UK regulatory environment. The Committee of Advertising Practice, the lead UK organisation that writes and maintains the UK advertising codes to ensure advertising in the UK is legal, decent, honest and truthful, has adopted the DH NPM in their recently published regulatory statement on food and soft drink advertising to children, which was developed following public consultation and will come into effect in the UK on 1 July 2017.[37]

Consumer organisations have set up campaigns in the UK and Australia asking supermarkets to remove unhealthy food and drink items from payment and queuing areas[38–40] but it is not known if such campaigns exist for retails outlets in hospitals. NICE quality standards are a set of prioritised statements, which draw on existing guidance to set out the priority areas for quality improvement in health and social care. They are designed to drive measurable quality improvements, yet, using them in the current study has proven challenging. Quality statement 1 refers to the availability of 'healthy food and drink options' in vending machines, yet nowhere in NICE quality standard 94 or in the associated documents is there a clear definition of what 'healthy' means, and it was on this basis that the DH NPM was chosen as a method of classification. Moreover, the definition of compliance to quality statement 1, where a vending machine contained both a healthy food and a healthy drink was chosen by the study authors given the lack of guidance on what constitutes as compliance from NICE quality standard 94. The authors considered compliance to quality statements 2 and 3 if most retail outlets in the hospitals adhered to the respective statement. 'Evidence' of 'arrangements to display healthy food and drink options in prominent places' constitutes adherence to quality statement 3 according to

NICE quality standard 94. However, challenges arose in assessing what constituted a 'healthy option' and a 'prominent place', and was subjective. The measurement of adherence to all three quality statements is open to interpretation, and assessment of adherence was a challenging task given their non-quantifiable and non-specific nature. NICE quality standard 94 would benefit from being measurable and, in the first instance, hospital venues could strive towards a balance of 50% healthy and 50% less healthy food and drinks.

The main limitation of the current study is that only two hospitals were included, both of which are in the same geographical area in England. Moreover, a second assessor would have provided an additional independent evaluation of the offerings. A further limitation is that the methodological approach for quality statements 2 and 3 is qualitative and subjective in nature, and future studies could use published tools which provide quantitative measurement variables and describe a systematic approach to data collection.[7 41] Further research is needed on a more representative sample of hospitals to fully understand compliance to NICE quality standard 94. However, such a study would be subject to similar limitations in terms of the vague nature of this quality standard and a more objective means of assessing adherence to the statements within the quality standard would be necessary.

In conclusion, the current study showed that two NHS hospitals demonstrated poor compliance with quality statements 1–3 in NICE quality standard 94. The lack of availability of healthy foods and drinks, the absence of nutritional information on the menus, the lack of advertising and display of healthy items and the consistent advertising and prominent display of unhealthy items highlights that improvements are required in NHS venues such as hospitals. These findings have been shared with the NHS trust of one of these hospitals and re-evaluation will take place in the future. However, as the prevalence of childhood obesity continues to rise globally, it is important that every opportunity is taken to improve the nutrition environment for children's food choices. Hospitals have a duty to provide consumers with the information required to make informed nutrition choices and should take the lead in supplying food and drinks that reflect evidence-based nutrition.[28]

**Acknowledgements** The authors thank Amanda Chong for assistance in the preparation of the manuscript and acknowledge support from the National Institute of Health Research (NIHR) Bristol Nutrition Biomedical Research Unit.

**Contributors** AJ and JPHS designed and collected the data. PF, SP and CB supervised and oversaw AJ during the data collection period. LB calculated and analysed the nutrient model scores. FEL, LB and ARN supervised the data analyses and prepared the manuscript. All authors revised the manuscript and approved the final version.

**Funding** National Institute of Health Research (NIHR) Bristol Nutrition Biomedical Research Unit.

**Disclaimer** This is an independent opinion from a Biomedical Research Unit (now Centre) in the National Institute for Health Research Biomedical Research Centre and Unit Funding Scheme. The views expressed in this publication are those of the authors and not necessarily those of the National Health Service, the National Institute for Health Research or the Department of Health.

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
