## [Reviewer comments · BMJ Open]

ARTICLE DETAILS

TITLE (PROVISIONAL)	Are food and drink retailers within NHS venues adhering to NICE Quality Standard 94 guidance on childhood obesity? A cross-sectional study of two large secondary care NHS hospitals in England
AUTHORS	James, Alice; Birch, Laura; Fletcher, Peter; Pearson, Sally; Boyce, Catherine; Ness, Andy; Shield, Julian; Lithander, Fiona

VERSION 1 – REVIEW

REVIEWER	Christina Vogel Senior Research Fellow University of Southampton United Kingdom
REVIEW RETURNED	07-Jul-2017

GENERAL COMMENTS	This manuscript describes assessment of the nutrition environment of retail outlets and restaurants in two hospitals in south west England. The manuscript is well written and the study takes a novel approach whereby assessing outlet nutrition environments against a NICE Quality Standard. There are a number of limitations with the study such as the lack of reliability assessment due to only having a single observer which have been mentioned by the authors. My primary concerns include the lack of theoretical underpinning and acknowledgment of existing consumer nutrition environment tools and the subjective nature of the assessment methods and description of the results for Quality Standard statements 2 & 3. I have listed major and minor comments below. Major comments: 1. The theoretical description of the consumer nutrition environment and knowledge or use of prior tools is missing from the introduction. The brief definition of the consumer nutrition environment in the introduction does not properly set up the study. A description of the factors that consumers encounter in venues where food products are purchased is missing and very little reference is made to prior literature. The authors provide a sentence about the school food environment however based on the Glanz model of nutrition environments, which the authors have cited, the school food environment is a separate environment from the consumer nutrition environment. An alternative body of literature that could be referred to in the literature is that concerning the concept of choice architecture and its associated typology (Hollands et al, 2013). The three statements of the Quality Standard refer to availability, nutrition labelling and placement which are well aligned with the choice architecture typology.
---

The authors claim that there are very few tools to measure the consumer nutrition environment however this is not entirely true. There are a large number of tools openly accessible on the US National Cancer Institute website that take an international perspective (<https://epi.grants.cancer.gov/mfe/instruments>). I feel the introduction would be enhanced if the theoretical underpinning of the project and the link to the Quality Standards were included.

2. As mentioned above, there are a number of consumer nutrition environment assessment tools available. I feel that the methods used to assess Quality Standards statement 1 are good and the detailed description enables readers to understand the process and justification for the approach. That particular measure is quantitative with clearly defined cut-off points, though my personal opinion is that they are quite generous but make a good starting place for incremental improvements toward a 'gold standard'. The methodological approach and description for statements 2 and 3 are considerably weaker and require improvement to be aligned with other consumer nutrition environment assessment tools. It is unclear why these two standards have not been assessed quantitatively. A quantitative measure of nutrition information and product shelf and store placement would have been possible and would enhance the quality of the assessment. The tool by Black et al, 2014 or the Gro Promo tool (Kerr et al, 2012), for example, could have been adapted for the two measures. Such tools provide quantitative measurement variables and describe a systematic approach to data collection. The data collection process for statements 2 and 3 could be more clearly described. I also wonder why the NPM was not used for grouping of food products in the placement assessment? It would be good if the authors were able to better describe and justify the assessment measures for Quality Standard statements 2 and 3. The descriptions of the results of these two measures are at times worded quantitatively could be reviewed.

3. The inclusion of price and price promotions does seem appropriate when price is not mentioned in NICE Quality Standard statements. The description of the price assessment and associated analysis could be enhanced. The justification for, description of and presentation of the results for the price variables could be improved or removed.

Minor comments:

1. It would be easier for the reader if sentences did not start with a number, particularly early in the introduction, and it would be consistent with small numbers - written out or numerical
2. Details of the Quality Standard statements are needed in the introduction or methods section as they first appear in the results section
3. Line 71, place commas after 'venue' and 'hospital'
4. Lines 92-96, please define each type of outlet with regards to food sold, size etc. Also describe whether the restaurants target staff more than patients or whether available to both groups
5. Line 94, the number of outlets don't sum to 7
6. The assessment tool subsection would be enhanced if the assessment methods for each section were described in their own paragraph in a manner similar to that of the results section which is nicely structured also Table 3 is difficult to follow in terms of reference to statement 2 or 3
7. The table order is not in order and not all Tables are referred to in the text

	8. Table 4 is very difficult to interpret based on the description of the assessment methods for statements 2 & 3 (as per point 2 above) 9. Line 178, remove ‘, where healthy.....or more,’ repeated on next line 10. Line 203-204, final sentence would be better placed in the methods section 11. Lines 306-311, this description may be better placed in the methods section 12. Lines 325-338, may be useful to add a recommendation for more detailed Quality Standard statements so they can be made measureable as suggested they should be in line 326. Perhaps a balance of healthy (50%) and unhealthy (50%) products would be recommended in the first instance. 13. The discussion only briefly mentions the challenges of improving the food environment within hospitals (i.e. financial benefits from commercial company rent in hospitals etc) and it would be good if these were discussed in more detail alongside the opportunities (i.e. contractual arrangements). Perhaps further qualitative work with hospital executives and commercial outlet managers is required to better understand these.
--	---

REVIEWER	Callie L Brown Wake Forest University
REVIEW RETURNED	19-Jul-2017

GENERAL COMMENTS	This manuscript by Ms. James et al describes a cross-sectional study assessing the healthfulness of food and drink options sold at restaurants, cafes, shops, and vending machines in two NHS hospitals in England. The authors found that adherence to Quality Statements 1-3 of the UK National Institute for Health and Care Excellence Quality Standard 94 was overall quite poor, with unhealthy foods served more frequently in vending machines, nutritional information was rarely available, and hospital shops and cafes prioritized display and advertisement of unhealthy foods. Overall this is a well-written paper about an important topic – hospitals and health clinics would do well to improve the healthfulness of the foods and drinks they offer for sale. However, I do have a few major concerns about the manuscript. The authors present a new assessment (C-NET) that was developed by the lead author for this study. A primary component of the tool is the UK Department of Health Nutrient Profiling Model (NPM), which is an existing tool that has been used previously although it is not clear if it has been validated in studies similar to this one or in the way that the authors used it (classifying foods as health vs. less healthy). Unfortunately the authors present no validation of this tool which makes it difficult to know how to interpret their results. Finally, all assessments were done by one researcher (the same researcher who developed the tool). I would considering reframing the paper such that the methods are not presented as the development of a novel tool, but instead focus on the existing measure that was used for quality statement 1 and then be more explicit about how the the questions in table 3 were developed. Minor point: throughout the manuscript please be consistent as to whether you place the reference before punctuation at the end of the sentence, or after.
---

Introduction:

- The first paragraph of the introduction sets up a paper to talk about interventions to prevent childhood obesity (and the methods section introduces child health services available at those facilities) yet the rest of the manuscript doesn't seem to apply strictly to children, so it is unclear why this was the initial focus. Do the NICE standards only apply to children? If not, I would refocus this away from a perspective of childhood obesity prevention.
- The authors state in 2 of the last 3 sentences of the introduction that few tools exist to assess the consumer nutrition environment of hospitals and that the authors will develop such a tool; however, there is no background presented about the types of tools that currently exist. The authors present 3 studies in the US and Canada that evaluated hospitals similarly – why were these existing tools not used for this study? More discussion as to why this is a gap in the literature is needed.

Methods:

- Line 92: Please clarify if the smaller hospital truly does not provide inpatient care or only does not provide inpatient paediatric care.
- Line 94: The authors state that there were 7 food retail outlets but then only list 6
- Line 105: Please explain why you chose NPM as your assessment tool of choice to evaluate the healthfulness of foods in vending machines. In the discussion the authors discuss some of the weaknesses of this tool (orange SSB and baked crisps being classified as healthy) and this seems to have a significant impact on the conclusions reached for Quality Statement 1. Are there other tools that would be have been more applicable?
- Line 149: What does displayed prominently mean? The authors have not provided sufficient detail for this study to be replicated by others.
- Line 157: The authors state that “interpretation of these data and the identification of themes was reviewed and discussed by multiple authors (AJ, LB, FEL) throughout the process to validate the findings.” It is not clear to me what the authors mean by this sentence, however, discussion among authors certainly does not equate to validation of findings. This part of the sentence should be removed.

Results:

- Lines 172-180: Please clarify that the data being presented here are *mean* NP score (SD)
- Lines 209: It is stated that adherence to quality statement 3 varied within the 2 restaurants, however, it isn't clear within this section which of these foods were considered healthy/unhealthy. The authors state later in the discussion that the pastry and cheese plate were considered unhealthy but this is not clear in the results section.
- Table 4 is not mentioned in the text of the results section, and this may help to clear up the confusion from the above point.
- Line 214: I would recommend splitting the cafes section into its own paragraph.

Discussion:

- Line 305: please describe how the NEMS is different from your tool
- Line 341: the authors are right to report that only having 2 hospitals from the same geographical area is a limitation of this study. However, I would be interested to know how the authors think these hospitals are similar or different from other hospitals in England.

VERSION 1 – AUTHOR RESPONSE

Reviewer 1:

There are a number of limitations with the study such as the lack of reliability assessment due to only having a single observer which have been mentioned by the authors. My primary concerns include the lack of theoretical underpinning and acknowledgment of existing consumer nutrition environment tools and the subjective nature of the assessment methods and description of the results for Quality Standard statements 2 & 3. I have listed major and minor comments below.

Reviewer 1 Major comment 1:

The theoretical description of the consumer nutrition environment and knowledge or use of prior tools is missing from the introduction. The brief definition of the consumer nutrition environment in the introduction does not properly set up the study. A description of the factors that consumers encounter in venues where food products are purchased is missing and very little reference is made to prior literature. The authors provide a sentence about the school food environment however based on the Glanz model of nutrition environments, which the authors have cited, the school food environment is a separate environment from the consumer nutrition environment.

Response from authors:

Thank you very much for these helpful comments. We have rewritten the Introduction and believe that it provides greater context for the current study. We have included a theoretical description of the consumer nutrition environment as you recommended as follows: "The consumer nutrition environment is one of four types of food environments and describes what an individual encounters when he or she enters a venue where food is purchased or consumed (Glanz et al, 2005)." In the updated Introduction, we have described the factors that consumers encounter in venues, as you suggested as follows: "The consumer nutrition environment accounts for factors that influence food choice within shops such as availability, price, promotions, placement, variety, quality and nutrition information of food and drink (Glanz et al., 2005; Black et al., 2014) and it is understood that consumers' dietary choices are affected by these factors (Hawkes et al., 2008 and Black et al., 2014). With respect to your comment on "acknowledgment of existing consumer nutrition environment tools" we have described the tools that are available in the literature, discussed the fact that many have not been validated and that most have been used solely in North America. We have described existing tools that assess consumer nutrition environment as those which either evaluate individual factors, (Mc Donald et al., 2006) or those which allowed the creation of a composite score of the environment such as the Hospital Nutrition Environment Scan (Winston et al., 2013, Winston et al., 2013), or the Nutritional Environmental Measured Survey-cafeteria (NEMS-C) (Lesser et al., 2012)". We have removed the sentence about the school food environment as the Reviewer correctly pointed out that this is a separate environment to the consumer nutrition environment.

- Black C, Ntani G, Inskip H, et al. Measuring the healthfulness of food retail stores: variations by store type and neighbourhood deprivation. *Int J Behav Nutr Phys Act.* 2014;11:69.
- Glanz K, Sallis JF, Saelens BE, et al. Healthy nutrition environments: concepts and measures *Am J Health Promot* 2005;19(5):330-3.
- Hawkes C: Dietary implications of supermarket development: a global perspective. *Dev Policy Rev* 2008, 26(6):657–692.
- McDonald CM, Karamlou T, Wengle JG, et al. Nutrition and exercise environment available to outpatients, visitors, and staff in Children's hospitals in Canada and the United States. *Arch Pediatr Adolesc Med.* 2006;160(9):900-5.
- Winston CP, Sallis JF, Swartz MD, et al. Reliability of the hospital nutrition environment scan for cafeterias, vending machines, and gift shops. *J Acad Nutr Diet.* 2013 ;113(8):1069-75.

- Winston CP, Sallis JF, Swartz MD, et al. Consumer nutrition environments of hospitals: an exploratory analysis using the hospital nutrition environment scan for cafeterias, vending machines, and gift shops, 2012. *Prev Chronic Dis.* 2013;10:E110.

Comment:

An alternative body of literature that could be referred to in the literature is that concerning the concept of choice architecture and its associated typology (Hollands et al, 2013). The three statements of the Quality Standard refer to availability, nutrition labelling and placement which are well aligned with the choice architecture typology.

Response from authors:

Thank you for this comment. The authors decided to introduce the concept of choice architecture in the Discussion and have added text as follows: “Choice architecture describes the concept that behaviour could be changed in anticipated ways by changing the environments where people make choices (Marteau et al., 2011; Hollands et al., 2013). The alteration of micro-environments, which are settings where people may congregate for a specific purpose (Swinburn et al., 1999), including hospitals, may be one approach to encourage healthier dietary behaviour, and studies have shown that minor changes in accessibility to food can decrease food intake (Rozin et al., 2011)”

- Marteau TM, Ogilvie D, Roland M, Suhrcke M, Kelly MP: Judging nudging: can nudging improve population health? *BMJ* 2011, 342:d228.
- Hollands GJ, Shemilt I, Marteau TM, Jebb SA, Kelly MP, Nakamura R, Suhrcke M, Ogilvie D. Altering micro-environments to change population health behaviour: towards an evidence base for choice architecture interventions. *BMC Public Health.* 2013 Dec 21;13:1218.
- Swinburn B, Egger G, Raza F: Dissecting obesogenic environments: the development and application of a framework for identifying and prioritizing environmental interventions for obesity. *Prev Med* 1999, 29(6):563–570.
- Rozin P, Scott S, Dingley M, Urbanek JK, Jiang H, Kaltenbach M: Nudge to nobesity I: minor changes in accessibility decrease food intake. *Judgment Decis Making* 2011, 6(4):323–332.

Comment:

The authors claim that there are very few tools to measure the consumer nutrition environment however this is not entirely true. There are a large number of tools openly accessible on the US National Cancer Institute website that take an international perspective (<https://epi.grants.cancer.gov/mfe/instruments>). I feel the introduction would be enhanced if the theoretical underpinning of the project and the link to the Quality Standards were included.

Response from authors:

Thank you for this comment. In the Introduction, we have included a paragraph on the tools that are currently available to measure the consumer nutrition environment. In this paragraph, we explain that regardless of the tool, few have been validated with respect to their measurement of association between the consumer nutrition environment and health, which has been described by Black et al., (2014). We have stated that “There are many other consumer nutrition environment tools in the literature that have been used in different venues which assess availability, price, variety and quality of fruit and vegetables, advertising, product placement, price promotions and labelling (Black et al., 2014; Gustafson et al., 2012 and Ohri-Vachaspati et al., 2010). Regardless of whether these tools assess individual factors or allow the creation of a composite score, it is noteworthy that few have been deemed reliable or valid (Black et al., 2014). Moreover, most tools have been developed and used in the United States and Canada and have not been used outside North America (Black et al., 2014).”

- Black C, Ntani G, Inskip H, et al. Measuring the healthfulness of food retail stores: variations by store type and neighbourhood deprivation. *Int J Behav Nutr Phys Act.* 2014;11:69.
- Gustafson A, Hankins S, Jilcott S: Measures of the consumer food store environment: a systematic review of the evidence 2000–2011. *J Community Health* 2012, 37(4):897–911.

• Ohri-Vachaspati P, Leviton LC: Measuring food environments: a guide to available instruments. *Am J Health Promot* 2010, 24(6):410–426.

Reviewer 1 Major comment 2:

As mentioned above, there are a number of consumer nutrition environment assessment tools available. I feel that the methods used to assess Quality Standards statement 1 are good and the detailed description enables readers to understand the process and justification for the approach. That particular measure is quantitative with clearly defined cut-off points, though my personal opinion is that they are quite generous but make a good starting place for incremental improvements toward a 'gold standard'.

Response from authors:

Thank you for your positive comments on the methods used to assess Quality Standard 1.

Comment:

The methodological approach and description for statements 2 and 3 are considerably weaker and require improvement to be aligned with other consumer nutrition environment assessment tools. It is unclear why these two standards have not been assessed quantitatively. A quantitative measure of nutrition information and product shelf and store placement would have been possible and would enhance the quality of the assessment. The tool by Black et al, 2014 or the Gro Promo tool (Kerr et al, 2012), for example, could have been adapted for the two measures. Such tools provide quantitative measurement variables and describe a systematic approach to data collection.

Response from authors:

Thank you for this comment. The Reviewer is quite right that Quality Standard 2 and 3 have not been assessed quantitatively and as such, this has been described as a further limitation in the Discussion. Please note that the subsequent comment from Reviewer 1 is also addressed here. The following text has been added to the Discussion: "A further limitation is that the methodological approach for Quality Statements 2 and 3 is qualitative and subjective in nature. Future studies could use published tools which provide quantitative measurement variables and describe a systematic approach to data collection (Kerr et al., 2012; Black et al., 2014) or use the NP model to group food products in the placement assessment."

• Black C, Ntani G, Inskip H, Cooper C, Cummins S, Moon G, Baird J. Measuring the healthfulness of food retail stores: variations by store type and neighbourhood deprivation. *Int J Behav Nutr Phys Act*. 2014 May 23;11:69

• Kerr J, Sallis JF, Bromby E, Glanz K: Assessing reliability and validity of the GroPromo audit tool for evaluation of grocery store marketing and promotional environments. *J Nutr Educ Behav* 2012, 44(6):597–603.

Comment:

The data collection process for statements 2 and 3 could be more clearly described. I also wonder why the NPM was not used for grouping of food products in the placement assessment? It would be good if the authors were able to better describe and justify the assessment measures for Quality Standard statements 2 and 3. The descriptions of the results of these two measures are at times worded quantitatively could be reviewed.

Response from authors:

The NPM was not used for grouping of food products in the placement assessment and whilst this is an excellent suggestion, we have noted this in the Discussion as a limitation as follows: "A further limitation is that the methodological approach for Quality Statements 2 and 3 is qualitative and subjective in nature. Future studies could use published tools which provide quantitative measurement variables and describe a systematic approach to data collection (Kerr et al., 2012; Black et al., 2014) or use the NP model to group food products in the placement assessment."

The assessment measures for Quality Statements 2 and 3 were designed according to the descriptions of NICE Quality Standard 94 (<https://www.nice.org.uk/guidance/qs94>). For example, in relation to Quality Statement 2, NICE Quality Standard 94 states that “Nutritional information....includes details on the calorie content of meals as well as information on the fat, saturated fat, salt and sugar content. If the nutritional value of recipes is not known, ingredients should be listed and cooking methods described. [Adapted from expert consensus and NICE guideline PH35, recommendation 8]”. The authors translated this information into the questions around the nutritional information that should be listed i.e. calories, fat, saturated fat and sugar. And the authors took it further whereby if such data were unavailable, ingredients should be listed and cooking methods described. This approach describes the origin of the assessment against Quality Statement 2.

The Quality Statement 3 includes phrases such as 'can easily find' healthy food and that these items were 'prominently displayed'. It should be noted that no further details were provided in the NICE Quality Standard 94 and that the guidance was open to interpretation by the authors. To help to clarify this in the Methods section, the relevant paragraph has been rewritten and now reads as follow: “To measure adherence to Quality Statements 2 and 3, she evaluated the quality of the consumer nutrition environment within each hospital restaurant, café or shop through answering a series of yes/no questions which centred around 3 main themes: the provision of nutrient information for meals on menus (Quality Statement 2), advertising and promotions (Quality Statement 3) and the prominent placing of healthy and unhealthy items in the retail outlets (Quality Statement 3) as outlined in Table 2. The themes were derived directly from the two Quality Statements. For assessment against Quality Statement 2, for example, the lead author examined the retail outlets for the display of calories, fat, saturated fat and sugar on menus, as specified in the Quality Statement itself. Quality Statement 2 was not relevant to the shops as they did not have menus. For Quality Statement 3, the lead author’s knowledge of the Eatwell Guide was used whereby foods within the food groups such as fruit, vegetables, dairy, wholegrain and high fibre foods, eggs, lean meat and pulses, were classified as healthy, and foods and drinks which are high in fat, salt and sugars such as biscuits, sweets, chocolate, and sugar sweetened beverages were classified as unhealthy. Quality Statement 3 includes phrases such as 'can easily find' healthy food, and that these items were 'prominently displayed' and descriptions of these terms are outlined in Table 2.”

Finally, in response to the Reviewer’s suggestion that ‘the descriptions of the results of these two measures are at times worded quantitatively could be reviewed’, the authors have considered this closely and feel that the descriptions of the results for Quality Statements 2 and 3 are written in a way that allows clarity. Each type of outlet, whether it is a shop, restaurant or café, is dealt with in a systematic approach to allow transparency of the results.

Comment:

The inclusion of price and price promotions does seem appropriate when price is not mentioned in NICE Quality Standard statements. The description of the price assessment and associated analysis could be enhanced. The justification for, description of and presentation of the results for the price variables could be improved or removed.

Response from authors:

Given that price is not mentioned in the NICE Quality Standard 94, the description of price has been removed from the Results section under Quality Statement 1.

The following two sentences however were not removed from the Results section under Quality Statement 3 as the authors felt that it provided context “There was a display in one shop with a sign which read ‘Big Kids Sweet Zone’ offering 35 different sweets all priced at less than £1. Both of the largest shops used posters to advertise a price deal on fresh fruit, however fruit stalls were not in a prominent location and were poorly stocked.”

Reviewer 1 minor comment 1:

1. It would be easier for the reader if sentences did not start with a number, particularly early in the introduction, and it would be consistent with small numbers - written out or numerical

Response from authors:

Thank you for this suggestion. The second sentence in the introduction now reads as follows: "Almost 10% of 4-5 year olds and 20% of 10-11 year olds in the UK are currently classified as obese".

Reviewer 1 minor comment 2:

2. Details of the Quality Standard statements are needed in the introduction or methods section as they first appear in the results section

Response from author:

Thank you for this comment. In the Introduction, we have described in greater details the NICE Quality Standard 94 whereby we have stated "The NICE Quality Standard 94 specifically refers to the availability of healthy foods and drinks in vending machines, nutrition labelling of menus and the prominent placement of healthy food and drink options"

Reviewer 1 minor comment 3:

1. Line 71, place commas after 'venue' and 'hospital'

Response from author:

This has been corrected

Reviewer 1 minor comment 4:

2. Lines 92-96, please define each type of outlet with regards to food sold, size etc. Also describe whether the restaurants target staff more than patients or whether available to both groups.

Response from author:

Thank you for this comment which helps the reader to further understand the characteristics of each outlet. The food that is sold in the restaurants and cafes is already described in the first paragraph of the Methods sections as follows: "The two restaurants served hot and cold meals throughout the day. The only meals served by the cafes were sandwiches and salads in addition to confectionary, savoury snack items, and hot drinks." The food sold in the shops has been added as follows: "All shops sold sweet and savoury items, cold drinks, and pre-packaged sandwiches and salads."

In terms of the accessibility and availability of the outlets to staff and patients, this is an important point and as such, the following sentences have been added to the first paragraphs of the Methods section: "All outlets were accessible to both staff and patients. The only two outlets which appeared to differentially target staff rather than patients were the two cafes, one on each hospital site, within the respective Education Centres where staff training and meetings take place and where several members of staff hold offices." Unfortunately, there is no information available on the size of each outlet.

Reviewer 1 minor comment 5:

3. Line 94, the number of outlets don't sum to 7

Response from author:

This has been corrected to 6 outlets.

Reviewer 1 minor comment 6:

6. The assessment tool subsection would be enhanced if the assessment methods for each section were described in their own paragraph in a manner similar to that of the results section which is nicely structured also Table 3 is difficult to follow in terms of reference to statement 2 or 3

Response from author:

The assessment tool subsection has been described in this way and the authors agree that this enhances the understanding of the methods used to assess Quality Statements 1, 2 and 3. Thank you for this comment.

Table 3 has been amended and an updated version submitted. The actual wording of the Quality Statements 2 and 3 has been added and it is the author's view that this table is now much easier to follow. In light of Reviewer 1's minor comment number 7, the tables have been renumbered and what was Table 3 is now Table 2.

Reviewer 1 minor comment 7:

7. The table order is not in order and not all Tables are referred to in the text

Response from author:

The name for Table 1 has remained unchanged. Table 2 has been renamed as Table 3 and Table 3 has been renamed as Table 2. The name for Table 4 has remained unchanged. The table that was not referenced in the text was Table 4 and it is now referenced in the Results section where the results for Quality Statement 3 are described.

Reviewer 1 minor comment 8:

8. Table 4 is very difficult to interpret based on the description of the assessment methods for statements 2 & 3 (as per point 2 above)

Response from author:

Table 4 has been updated whereby Quality Statement 3 has been added to the first row of the table as follows: "Quality Statement 3: "Children and young people, and their parents or carers, see healthy food and drink choices displayed prominently in local authority and NHS venues".

Reviewer 1 minor comment 9:

9. Line 178, remove ' , where healthy.....or more,' repeated on next line

Response from author:

This has been removed.

Reviewer 1 minor comment 10:

10. Line 203-204, final sentence would be better placed in the methods section

Response from author:

This sentence has been moved to the Methods Section.

Reviewer 1 minor comment 11:

11. Lines 306-311, this description may be better placed in the methods section

Response from author:

This description has been moved to the methods section to the subsection entitled Assessment Tool under the heading of Quality Statement 1.

Reviewer 1 minor comment 12:

12. Lines 325-338, may be useful to add a recommendation for more detailed Quality Standard statements so they can be made measurable as suggested they should be in line 326. Perhaps a balance of healthy (50%) and unhealthy (50%) products would be recommended in the first instance.

Response from author:

The following sentence has been added as a recommendation “NICE Quality Standard 94 would benefit from being measurable and, in the first instance, hospital venues could strive towards a balance of 50% healthy and 50% less healthy food and drinks available.”

Reviewer 1 minor comment 13:

13. The discussion only briefly mentions the challenges of improving the food environment within hospitals (i.e. financial benefits from commercial company rent in hospitals etc) and it would be good if these were discussed in more detail alongside the opportunities (i.e. contractual arrangements). Perhaps further qualitative work with hospital executives and commercial outlet managers is required to better understand these.

Response from author:

Thank you for this comment. The authors are content with the brief mention of the challenges of improving the food environment within hospitals that is currently in the manuscript. Given that the main aim of the study was not to consider these challenges and opportunities, the authors have decided not to expand on this.

Reviewer 2 Name: Dr Callie L Brown

This manuscript by Ms. James et al describes a cross-sectional study assessing the healthfulness of food and drink options sold at restaurants, cafes, shops, and vending machines in two NHS hospitals in England. The authors found that adherence to Quality Statements 1-3 of the UK National Institute for Health and Care Excellence Quality Standard 94 was overall quite poor, with unhealthy foods served more frequently in vending machines, nutritional information was rarely available, and hospital shops and cafes prioritized display and advertisement of unhealthy foods. Overall this is a well-written paper about an important topic – hospitals and health clinics would do well to improve the healthfulness of the foods and drinks they offer for sale. However, I do have a few major concerns about the manuscript.

Response from authors:

Thank you, Dr Brown, for your comprehensive review of this manuscript. We agree that this is an important topic and we believe that we address a gap in the literature where we understand that this is the first study to assess the consumer nutrition environment in hospitals in the UK.

Comment:

The authors present a new assessment (C-NET) that was developed by the lead author for this study. A primary component of the tool is the UK Department of Health Nutrient Profiling Model (NPM), which is an existing tool that has been used previously although it is not clear if it has been validated in studies similar to this one or in the way that the authors used it (classifying foods as health vs. less healthy).

Response from authors:

The UK Department of Health Nutrient Profiling Model has been used to define ‘healthy’ and ‘unhealthy’ foods and drinks for TV advertising to children and is used in the UK for this purpose. The British Heart Foundation Health Promotion Research Group has published a series of papers relating to the development of the model and its validation and these publications include the following:

- Rayner M, Scarborough P, Williams C. The origin of Guideline Daily Amounts and the Food Standards Agency's guidance on what counts as 'a lot' and 'a little'. *Public Health Nutrition* 2003; 7 (4); 549-556.
- Scarborough P, Rayner M, Stockley L. Developing nutrient profile models: a systematic approach. *Public Health Nutrition* 2007; 10; 330-336.
- Scarborough P, Rayner M, Stockley , Black A. Nutrition professionals' perception of the 'healthiness' of individual foods, *Public Health Nutrition* 2007; 10; 346-353.
- Scarborough P, Boxer A, Rayner M, Stockley L. Testing nutrient profile models using data from a survey of nutrition professionals, *Public Health Nutrition* 2007; 10; 337-345.
- Arambepola C, Scarborough M, Rayner M. Validating a nutrient profile model, *Public Health Nutrition* 2008; 11; 371–378.
- Arambepola C, Scarborough P, Boxer A, Rayner M. Defining 'low in fat' and 'high in fat' when applied to a food. *Public Health Nutrition* 2009; 12; 341-350.

The authors can confirm that the NPM has not been used to assess the consumer nutrition environment in hospitals in the UK because to the best of our knowledge, there are no studies in the literature which have assessed the consumer nutrition environment in UK hospitals.

Further details of the NPM itself, including the history of the model, the effectiveness of the model at differentiating foods on the basis of their nutrient composition, and other papers which have been published in peer reviewed international journal can be found here:
<https://www.ndph.ox.ac.uk/cpnp/files/about/uk-of-com-nutrient-profile-model.pdf> This reference is listed in the Bibliography of the current manuscript.

Comment:

Unfortunately, the authors present no validation of this tool which makes it difficult to know how to interpret their results.

Response from authors:

The authors agree with this comment. However, it is recognised in the literature that few tools that have been used to assess the consumer nutrition environment have been deemed valid or indeed provide the information required to measure the association between the food environments and diet (Black et al. 2014; Gustafson et al. 2012; Rimkus et al. 2013; McKinnon et al. 2009). The authors have included a sentence on this point in the Introduction of the current manuscript.

- Black C, Ntani G, Inskip H, Cooper C, Cummins S, Moon G, Baird J. Measuring the healthfulness of food retail stores: variations by store type and neighbourhood deprivation. *Int J Behav Nutr Phys Act.* 2014 May 23;11:69
- Gustafson A, Hankins S, Jilcott S: Measures of the consumer food store environment: a systematic review of the evidence 2000–2011. *J Community Health* 2012, 37(4):897–911.
- Rimkus L, Powell LM, Zenk SN, Han E, Ohri-Vachaspati P, Pugach O, Barker DC, Resnick EA, Quinn CM, Myllyluoma J, Chaloupka FJ: Development and reliability testing of a food store observation form. *J Nutr Educ Behav* 2013, 45(6):540–548
- McKinnon RA, Reedy J, Morrissette MA, Lytle LA, Yaroch AL: Measures of the food environment: a compilation of the literature, 1990–2007. *Am J Prev Med* 2009, 36(4 Suppl):S124–S133.

Comment:

Finally, all assessments were done by one researcher (the same researcher who developed the tool). I would considering reframing the paper such that the methods are not presented as the development of a novel tool, but instead focus on the existing measure that was used for quality statement 1 and then be more explicit about how the questions in table 3 were developed. Minor point: throughout the manuscript please be consistent as to whether you place the reference before punctuation at the end of the sentence, or after.

Response from authors:

Reviewer 2 is quite right in that all assessments were carried out by the lead author. We are in agreement that this is a limitation of this study and the authors have commented on this limitation in the Discussion whereby the following sentence is included 'a second assessor would have provided an additional independent evaluation of the offerings'. In addition, it is included in the section entitled 'Strengths and Limitation of the Study'.

The authors have thought carefully about your suggestion of reframing the paper. Following your recommendation, we have added to the methodology relating to the Nutrient Profiling Model, which is the measure that was used to assess compliance to Quality Statement 1. The text that we added here was in fact moved from the Discussion as it was deemed more appropriate to place it in the Methods section. This text reads as follows: "The NPM was developed by the Food Standards Agency in 2004-2005, and was subject to rigorous scientific scrutiny, extensive consultation, and review. It is supported by the independent Scientific Advisory Committee on Nutrition and a wide range of nutrition experts. The NPM was introduced as mandatory in 2007 by the UK Office of Communications and DH to regulate food and drinks in the context of television advertising to children."

Thank you for your suggestion to 'be more explicit about how the questions in table 3 were developed'. Reviewer 1 also commented on this point. We were unsure if you have access to the authors' responses to Reviewer 1's comments, so I have pasted the response that I gave to Reviewer 1 here (for your information, Table 3 had been relabelled as Table 2):

"The assessment measures for Quality Statements 2 and 3 were designed according to the descriptions of NICE Quality Standard 94 (<https://www.nice.org.uk/guidance/qs94>). For example, in relation to Quality Statement 2, NICE Quality Standard 94 states that "Nutritional information....includes details on the calorie content of meals as well as information on the fat, saturated fat, salt and sugar content. If the nutritional value of recipes is not known, ingredients should be listed and cooking methods described. [Adapted from expert consensus and NICE guideline PH35, recommendation 8]". The authors translated this information into the questions around the nutritional information that should be listed i.e. calories, fat, saturated fat and sugar. And the authors took it further whereby if such data were unavailable, ingredients should be listed and cooking methods described. This approach describes the origin of the assessment against Quality Statement 2.

The Quality Statement 3 includes phrases such as 'can easily find' healthy food and that these items were 'prominently displayed'. It should be noted that no further details were provided in the NICE Quality Standard 94 and that the guidance was open to interpretation by the authors. To help to clarify this in the Methods section, the relevant paragraph has been rewritten and now reads as follow: "To measure adherence to Quality Statements 2 and 3, she evaluated the quality of the consumer nutrition environment within each hospital restaurant, café or shop through answering a series of yes/no questions which centred around 3 main themes: the provision of nutrient information for meals on menus (Quality Statement 2), advertising and promotions (Quality Statement 3) and the prominent placing of healthy and unhealthy items in the retail outlets (Quality Statement 3) as outlined in Table

2. The themes were derived directly from the two Quality Statements. For assessment against Quality Statement 2, for example, the lead author examined the retail outlets for the display of calories, fat, saturated fat and sugar on menus, as specified in the Quality Statement itself. Quality Statement 2 was not relevant to the shops as they did not have menus. For Quality Statement 3, the lead author's knowledge of the Eatwell Guide was used whereby foods within the food groups such as fruit, vegetables, dairy, wholegrain and high fibre foods, eggs, lean meat and pulses, were classified as healthy, and foods and drinks which are high in fat, salt and sugars such as biscuits, sweets, chocolate, and sugar sweetened beverages were classified as unhealthy. Quality Statement 3 includes phrases such as 'can easily find' healthy food, and that these items were 'prominently displayed' and descriptions of these terms are outlined in Table 2."

Thank you for your minor point; I can confirm that we have corrected the manuscript for consistency between the reference and the punctuation.

Comment:

Introduction

- The first paragraph of the introduction sets up a paper to talk about interventions to prevent childhood obesity (and the methods section introduces child health services available at those facilities) yet the rest of the manuscript doesn't seem to apply strictly to children, so it is unclear why this was the initial focus. Do the NICE standards only apply to children? If not, I would refocus this away from a perspective of childhood obesity prevention.

Response from authors:

The NICE Quality Standard 94 apply specifically to children. The Standard is entitled "Obesity: prevention and lifestyle weight management in children and young people" and according to NICE, this quality standard covers a range of approaches at a population level to prevent children and young people aged under 18 years from becoming overweight or obese (<http://www.maternal-and-early-years.org.uk/nice-quality-standard-94-obesity-prevention-and-lifestyle-weight-management-in-children-and-young-people>)

Comment:

- The authors state in 2 of the last 3 sentences of the introduction that few tools exist to assess the consumer nutrition environment of hospitals and that the authors will develop such a tool; however, there is no background presented about the types of tools that currently exist. The authors present 3 studies in the US and Canada that evaluated hospitals similarly – why were these existing tools not used for this study? More discussion as to why this is a gap in the literature is needed.

Response from authors:

Thank you for this comment about tools which was also raised by Reviewer 1. The authors have now added background information to the Introduction about the other types of tools that current exist. The tools that were used in the three studies mentioned by Reviewer 2 that were carried out in the US and Canada were specific to North America. In addition, they were not designed to measure compliance against NICE Quality Standards, which are specific to the UK.

The current study addresses a gap in the literature where a comprehensive tool was developed to measure the healthfulness of foods and drinks available in outlets in UK NHS Hospitals, and to assess these against NICE Quality Standard 94. Based on your comment, we have added a sentence at the end of the Introduction which we hope clarifies to the reader what this gap in the literature is. The sentence reads as follows "The current study addresses a gap in the literature by developing a consumer nutrition environment tool to measure the healthfulness of food outlets of NHS hospitals, and assesses this against NICE Quality Standard 94 which refers to the availability of healthy foods and drinks in vending machines, nutrition labelling of menus, and the prominent placement of healthy food and drink options."

Comment:

Methods:

- Line 92: Please clarify if the smaller hospital truly does not provide inpatient care or only does not provide inpatient paediatric care.

Response from authors:

Thank you for pointing out this point of ambiguity. We have clarified the following sentence in the Methods section: "Both hospitals offer adult inpatient care; paediatric, day-time access for assessment and out-patients are available in both, but only the larger hospital provides paediatric inpatient care".

Comment:

- Line 94: The authors state that there were 7 food retail outlets but then only list 6

Response from authors:

This has been corrected to 6.

Comment:

- Line 105: Please explain why you chose NPM as your assessment tool of choice to evaluate the healthfulness of foods in vending machines.

Response from authors:

The nutrient profiling model was developed by the UK Food Standards Agency (FSA) in 2004-2005 as a tool to help Ofcom (UK media and communications regulator) to differentiate foods and improve the balance of television advertising to children (<https://www.gov.uk/government/publications/the-nutrient-profiling-model>). The NP model was subject to rigorous scientific scrutiny, extensive consultation and review. It is supported by the independent Scientific Advisory Committee on Nutrition (SACN) and a wide range of nutrition experts (<https://www.gov.uk/government/publications/the-nutrient-profiling-model>)

The NPM was chosen for three reasons:

- Given that Quality Statement 1 refers to the availability of 'healthy food and drink options' in vending machines, yet nowhere in NICE Quality Standard 94 is there a definition of what healthy means, the authors had no option but to choose a tool to differentiate 'healthy' from 'less healthy' for use in the current study.
- The NPM tool is widely regarded as scientifically robust and effective in identifying 'healthy' and 'less healthy' options
- The NPM tool is well established in the UK

There are other government endorsed NPMs available but, as outlined in the Discussion section of this manuscript, "The Committee of Advertising Practice (CAP), the lead UK organisation that write and maintain the UK Advertising Codes to ensure advertising in the UK is legal, decent, honest and truthful, have adopted the Nutrient Profiling Model in their recently published regulatory statement on food and soft drink advertising to children, which was developed following public consultation and will come into effect in the UK on 1 July 2017 (Committee of Advertising Practice (CAP). CAP

Consultation: food and soft drink advertising to children. 2016.

<https://www.asa.org.uk/asset/98337008-FA03-481B-92392CB3487720A8/>".

Comment:

In the discussion, the authors discuss some of the weaknesses of this tool (orange SSB and baked crisps being classified as healthy) and this seems to have a significant impact on the conclusions reached for Quality Statement 1. Are there other tools that would be have been more applicable?

Response from authors:

The authors agree that there are some weaknesses associated with the tool. Unfortunately however, there are no other tools that would have been more applicable for the assessment of the consumer nutrition environment in UK NHS hospitals. In addition, the nutrient profile model was tested on a range of foods in UK national databases (<https://www.ndph.ox.ac.uk/cnpn/files/about/uk-ofcom-nutrient-profile-model.pdf>). As such, any tools that were developed outside the UK could not have been used in the current study as they have not been tested using UK national food databases.

Comment:

- Line 149: What does displayed prominently mean? The authors have not provided sufficient detail for this study to be replicated by others.

Response from authors:

The following text was added to the Methods section, under the subheading 'Procedures' and provides further detail in the meaning of 'displayed prominently'. "Items were deemed to be prominently displayed if they were placed at the entrance to the retail outlet and hence visible to those walking past, beside queuing and payment areas, or if they were accompanied by signs advertising the product or detailing price promotions. Though price was not specifically mentioned in Quality Statements 1, 2 or 3, the price of vending machine items and the existence of price promotions were both assessed as part of this study. Quality Statement 1 asks that venues ensure that children and their carers 'can buy' healthy items and Quality Statement 3 requires that healthy food and drink items are 'promoted'. Both Statements thus require reasonable accessibility to healthy options. Accessibility applies not only to the physical display and number of items, but also to the financial accessibility."

Comment:

- Line 157: The authors state that "interpretation of these data and the identification of themes was reviewed and discussed by multiple authors (AJ, LB, FEL) throughout the process to validate the findings." It is not clear to me what the authors mean by this sentence, however, discussion among authors certainly does not equate to validation of findings. This part of the sentence should be removed.

Response from authors:

Thank you for this insightful comment. The words 'to validate the findings' have been removed.

Comment:

Results: • Lines 172-180: Please clarify that the data being presented here are *mean* NP score (SD)

Response from authors:

Thank you for this point. The text now reads: "When the 40 foods alone were assessed using the NPM, the mean (sd) NP score was 18.3, (9.3). Only 4 of the 40 (10%) foods were classified as healthy with a NP score of less than 4; these items were baked crisps, a packet of dried fruit and nuts, a packet of raw nuts and a muesli bar (mean (sd) NP score -1.3 (4.5)). When the other 36 (90%) food items were analysed according to the NPM, the mean (sd) NP score was 20.5 (sd 6.8), significantly greater than the cut-off for the less healthy classification of 4 points or more. The 4 healthy foods items were priced equivalently to similar less healthy items available in the vending machines.

Comment:

- Lines 209: It is stated that adherence to quality statement 3 varied within the 2 restaurants, however, it isn't clear within this section which of these foods were considered healthy/unhealthy. The authors state later in the discussion that the pastry and cheese plate were considered unhealthy but this is not clear in the results section.

Response from authors:

To help clarify this, 'Table 4' has been added to the end of this sentence in the Results section. It is hoped that this will draw the reader to examine Table 4, which documents the observed activities which shops, cafes and restaurants engaged in against which compliance to Quality Statement 3 was measured.

In addition, in the Results, after the statement of pastry and cheese plate, the following text was added 'both of which are less healthy options.'

Comment:

- Table 4 is not mentioned in the text of the results section, and this may help to clear up the confusion from the above point.

Response from authors:

Thank you. Table 4 has been added to the text of the Results section as you suggested.

Comment:

- Line 214: I would recommend splitting the cafes section into its own paragraph.

Response from authors:

Thank you for this suggestion; it is much clearer now.

Comment:

Discussion:

- Line 305: please describe how the NEMS is different from your tool

Response from authors:

There are a variety of Nutrition Environment Measures Scale (NEMS) tools available. These include the NEMS-Restaurant (NEMS-R), NEMS-Stores (NEMS-S), NEMS-Restaurants (NEMS-R) In their study of food offerings in Californian children's hospitals, Lesser et al (2012) modified the NEMS-R tool and named it the NEMS-C tool to make it applicable to all hospital food service venues.

The NEMS-R tool is the most widely used. It comprises a 25-item, checklist that requires systematic observation, and further details can be found in the original publication and online (Just, 2007).

The NEMS tools were unsuitable for the current study as they were designed in the US, and have been used in North America and as such are unsuitable for the UK context.

- Lesser LI, Hunnes DE, Reyes P, et al. Assessment of food offerings and marketing strategies in the food-service venues at California Children's Hospitals. *Acad Pediatr.* 2012;12(1):62-7.

- Just D. Could behavioral economics help improve diet quality for nutrition assistance program participants? *Economic Research Report.* 2007;43.

Comment:

- Line 341: the authors are right to report that only having 2 hospitals from the same geographical area is a limitation of this study. However, I would be interested to know how the authors think these hospitals are similar or different from other hospitals in England.

Response from authors:

It is the author's view that these hospitals are similar to other District General Hospitals in the UK in terms of size, capacity and speciality.

VERSION 2 – REVIEW

REVIEWER	Christina Vogel University of Southampton, United Kingdom
REVIEW RETURNED	09-Oct-2017
GENERAL COMMENTS	The authors have thoughtfully considered and addressed my comments. In my opinion, the manuscript is now suitable for publication. Congratulations to the authors on the invitation to present this work.